# Frailty Intervention through Nutrition Education and Exercise (FINE). A Health Promotion Intervention to Prevent Frailty and Improve Frailty Status among Pre-Frail Elderly—A Study Protocol of a Cluster Randomized Controlled Trial

**DOI:** 10.3390/nu12092758

**Published:** 2020-09-10

**Authors:** Nurul Izzati Mohd Suffian, Siti Nur ‘Asyura Adznam, Hazizi Abu Saad, Yoke Mun Chan, Zuriati Ibrahim, Noraida Omar, Muhammad Faizal Murat

**Affiliations:** 1Department of Nutrition and Dietetics, Faculty of Medicine and Health Sciences, Universiti Putra Malaysia, Serdang, Selangor 43400, Malaysia; nurulizzatisuffian@gmail.com (N.I.M.S.); hazizi@upm.edu.my (H.A.S.); cym@upm.edu.my (Y.M.C.); zuriatiib@upm.edu.my (Z.I.); noraidaomar@upm.edu.my (N.O.); muhammadfaizalmurat@gmail.com (M.F.M.); 2Malaysian Research Institute of Ageing, (My Ageing) Universiti Putra Malaysia, Serdang, Selangor 43400, Malaysia; 3Sports Academy, Universiti Putra Malaysia, Serdang, Selangor 43400, Malaysia

**Keywords:** community-dwelling, elderly, exercise, frailty, intervention, multi-component, nutrition education, randomized controlled trial

## Abstract

The ageing process has been associated with various geriatric issues including frailty. Without early prevention, frailty may cause multiple adverse outcomes. However, it potentially may be reversed with appropriate interventions. The aim of the study is to assess the effectiveness of nutritional education and exercise intervention to prevent frailty among the elderly. A 3-month, single-blind, two-armed, cluster randomized controlled trial of the frailty intervention program among Malaysian pre-frail elderly will be conducted. A minimum of total 60 eligible respondents from 8 clusters (flats) of Program Perumahan Rakyat (PPR) flats will be recruited and randomized to the intervention and control arm. The intervention group will receive a nutritional education and a low to moderate multi-component exercise program. To date, this is the first intervention study that specifically targets both the degree of frailty and an improvement in the outcomes of frailty using both nutritional education and exercise interventions among Malaysian pre-frail elderly. If the study is shown to be effective, there are major potential benefits to older population in terms of preventing transition to frailty. The findings from this trial will potentially provide valuable evidence and serve as a model for similar future interventions designed for elderly Malaysians in the community.

## 1. Introduction

A developing country such as Malaysia is not exempt from the fast changing demographic patterns in society and is expected to achieve the status of an ageing country by 2035 [1,2]. In line with this, the ageing process has been associated with various geriatric issues including frailty [3]. Frailty is a term commonly used by health care professionals to describe the condition of an older person who has chronic health problems, has lost functional abilities, and is likely to deteriorate further [4]. Frailty is a continuum process, and researchers commonly classify frailty into three categories (normal, pre-frail, and frail) according to the degree of deterioration [5]. Interestingly, it is not unidirectional but a dynamic and reversible process [6].

Several risk factors lead to frailty including socio-demographic status (advanced age, female, low educational, and socio-economic status), chronic disease, malnutrition, physical inactivity, cognitive impairment, poor functional status, and history of falls [2,7,8,9]. A clear relationship is also shown with adverse outcomes, such as disability/dependency, hospitalization, institutionalization, falls, poor mobility, depression, and death in community-dwelling older adults [5,10,11]. While those in an intermediate stage of frailty (pre-frail) present an increased risk of becoming frail within just three years [5].

The recent review on the global prevalence of pre-frail and frail elderly in the community ranged from 34.6% to 50.9% and 4.9% to 27.3%, respectively [12]. Asian countries recorded a higher prevalence range of pre-frailty (40% to 72%) and frailty (5% to 28%) than the global range [13,14]. Consistent with the worldwide trend, the frailty problem among Malaysian elderly has also emerged as a cause for concern [2]. In a large scale study conducted among community-dwelling elderly in an urban area in Malaysia, it was found that pre-frailty and frailty affected 61.7% and 8.9% of the older adults, respectively [15]. In addition, on the east coast of the peninsula, there are about 18.3% frail elders [16], and in Perak and Kelantan, the frailty affected 23.0% of the older adults [17].

Although the prevalence of the pre-frail elderly population is already at an alarming state, only a few interventions have focused on the pre-frail elderly population [18,19,20]. Pre-frail individuals have more than twice the risk of becoming frail compared to non-frail people [5]. However, because frailty appears to be a dynamic process and potentially reversible, implementing interventions for pre-frail elderly may prevent the development of frailty. Evidence shows that pre-frail elderly may respond better to interventions than elderly who have already moved to a frail state [21], probably due to less disability than that of the frail elderly [5]. Thus, there is potential for more intensive interventions among pre-frail elderly.

Unfortunately, research to improve frailty outcomes for frail elderly is still in its infancy, especially in Asian countries. Despite several nutritional interventions that were applied in a previous study, few convincing effects in improving frailty outcomes were apparent. Moreover, most of the studies involved supplementation and meal support, which are obviously costly [22,23,24,25] and hence run the risk of poor sustainability [3]. Even though there are limited interventions on nutrition education that have been conducted among frail elders but few studies able to report positive outcomes [26,27]. It includes the education on healthy food choices and dietary habit change [27,28].

In contrast, the evidence concerning the effectiveness of exercise intervention on the frailty outcome was undoubtedly more convincing [29,30,31,32]. In fact, exercise seems to play an essential role in any combination interventions, whereby additional intervention (e.g., nutritional intervention) can only lead to further improvement [33]. A recent systematic review suggests the importance of a combination intervention as these tend to be more effective than a single intervention [33], especially when diet and exercise are both included [33,34]. It has been shown that the characteristics of exercise programs that seem to result in better outcomes should include multicomponent training that is performed 3 days per week, with a duration of 3 months or longer and less than 60 min per session [35,36]. However, uncertainty still exists with regard to which exercise characteristics (type, frequency, and duration) are the most effective.

Evidence shows that elderly people living in rural areas have an economic disadvantage [37], which may put them at risk of being malnourished [38] and thus at high risk of being frail [39]. However, less is known about the poor elderly population living in urban areas such as the Program Perumahan Rakyat (PPR) flats in Kuala Lumpur area. In the 2000 census, Kuala Lumpur reached the status of 100 percent urban population [40]. Meanwhile, the PPR flats are low-cost public subsidized high-rise flats (5 to 18 floors) built for the resettlement of squatters and to provide housing to the economically disadvantaged individuals. According to Loh et al. [1], the PPR flats are places where the urban poor and a growing aging population reside. Thus, probably, the elderly who are living in the PPR flats are also at risk of being frail.

To the best of knowledge, there is no published frailty intervention that specifically targets both the degree of frailty and an improvement in the outcome using a combination of nutrition education and exercise interventions among Malaysian pre-frail elderly. Although a recent trial was conducted among the pre-frail elderly in Malaysia [20], the study only included single intervention and the focus was on the supplementation. Even though the supplementation given has the potential to reverse frailty and its outcomes, the efficacy remains doubtful. To address this gap, this study aims to develop, implement, and evaluate the effectiveness of a Frailty Intervention through Nutrition Education and Exercise (FINE) intervention program targeted at both improving the degree of frailty and the outcomes by using a combination of nutrition education and exercise interventions, and compare it with the general health education among Malaysian pre-frail elderly in a poor urban setting.

## 2. Materials and Methods

This study is designed and will be conducted and reported in keeping with the Consolidation Standards of Reporting (CONSORT) 2010 statement and its extension to cluster randomized trials [41].

### 2.1. Trial Design

The “FINE” project is a 3-month (12 weeks) intervention program, single-blind, two-armed cluster randomized controlled trial (cluster RCT), conducted among pre-frail elderly with a pre- and post-intervention and 3-month follow-up assessments to assess the effectiveness of the program. The trial mainly comprises nutritional education and exercise intervention among older people aged 60 years and above in the PPR flats in Kuala Lumpur. The trial is registered prospectively at ClinicalTrial.gov with registration number NCT04327544 on 30 March 2020.

### 2.2. Participants

#### 2.2.1. Ethics Approval

The clinical trial is conducted in accord with the guidelines of the Declaration of Helsinki and the guidelines of Good Clinical Practice (GCP). Written informed consent will be sought from all the study participants prior to the commencement of the study. The study already has been approved by the Ethics Committee for Research involving Human Subjects Universiti Putra Malaysia (JKEUPM) (JKEUPM-2019-335). Permission for field data collection in the PPR flats has already been attained from the Dewan Bandaraya Kuala Lumpur (DBKL), as well as the Head Officer of the PPR flats in Kuala Lumpur.

#### 2.2.2. Screening of Frailty Status

In total, there are 32 PPR flats listed under Kuala Lumpur area (DBKL, 2019). One of the PPR flats is excluded for the purpose of acceptability study as part of the development process of education materials, while the remaining 31 flats are divided into four different zones. Two zones with the highest prevalence of pre-frail elderly will be selected for the intervention study. However, due to no previous data having been recorded regarding the frailty status of the elderly, a cross-sectional study will be conducted to assess the current frailty status of the elderly. Using the one proportion formula by Daniel [42] with estimated prevalence of pre-frailty, 72.4% [14] and additional 20% subjects, the number of samples included in the screening study is 368.

Meanwhile, the minimum required cluster recruited from each zone is based on the calculation in the actual intervention study which is 4. Since there are 4 zones, in total 16 flats (cluster) will be included and randomly selected. Note that a cluster refers to one PPR building that has at least 50 residents aged 60 years and above. A proportionate sampling technique will be used for the recruitment of the respondents from each PPR to ensure that the total number of respondents will fairly represent each PPR flats. Frailty status will be assessed using Malay language standardized phenotype of frailty questionnaire [5]. The frailty phenotype evaluates five components of the frailty syndrome and allocates one point for each criterion met; respondents meeting zero criteria are defined as non-frail, whereas those meeting one or two criteria are defined as pre-frail, and those meeting three, four, or five criteria are defined as frail.

#### 2.2.3. Eligibility and Recruitment

Based on the result from the screening, two zones with the highest number of pre-frail elderly will be selected. Pre-frail people 60 years and older from the selected PPR flats will be invited to participate in this FINE program. After the invitation process, the information sheets will be distributed and explained. Those individuals that agree to participate will be requested to provide written informed consent, and subsequently, they will be further screened to identify their eligibility to participate. The inclusion and exclusion criteria are as in Table 1. Once individuals have been screened as being eligible to participate, they will be invited to attend a pre-intervention data collection (baseline assessment). The overview of the frailty intervention study flow is shown in Figure 1.

#### 2.2.4. Sample Size Calculation

The sample size for each arm was calculated by using two group mean comparison formula [25]. Both the mean and standard deviation from the study conducted by Tarazona-Santabalbina et al. [29] in Spain was used as the reference, and based on the calculation using the result for the variable of the Short Physical Performance Battery (SPPB), the sample size required for each intervention and control arm is 15 respondents.

With a hypothesized intra-cluster correlation coefficient (ICC) 0.02 [1], fixed cluster size of 4 [45], and a minimum of 15 respondents required under individual randomization, the calculated minimum required cluster size and sample size is 4 and 16 per arm, respectively. After an adjustment for the estimated response rate, a sample size of 30 respondents per arm is required to provide of 80%, with α = 0.05. In total, the respondents for both arms are 60 elderlies with approximately 8 respondents from each cluster (4 clusters for each arm).

#### 2.2.5. Randomization and Allocation

Randomization will be conducted after the baseline assessment is completed. Randomization takes place at the zone level to reduce the risk of exposure of the control arm to the intervention effect. PPR flats with the highest prevalence of pre-frail elderly in two zones resulting from the screening will be randomized into the intervention arm and the control. Using a computer-generated random number sequence, the study statistician will allocate the PPR flats into the intervention and control group based on their zones.

### 2.3. Intervention

Prior to the commencement of the FINE program, a process of development of the frailty intervention module and education materials will be conducted. The educational materials will be developed based on the frailty intervention module to serve as a reference or reading materials for the respondents throughout the program. It includes flipchart, PowerPoint slides, booklet, and posters. During a 12-week (3-month) intervention period, in overall, participants attend one session of frailty, exercise, and healthy eating talk; 20 sessions of a low to moderate intensity multi-component exercise course; and 5 sessions of nutrition education intervention. Additionally, there is one session during the following 3 months after the intervention ends. It should be noted that each session will be held in the facility area or hall of the PPR flats for around 60 min.

As an introduction, a few talk sessions and an exercise demonstration will be conducted for few days of the first week regarding frailty, exercise, and nutrition with the aims of introducing, refreshing, and improving the knowledge regarding the topic. Each topic will be delivered by the research team based on the frailty intervention module. In the following week, the exercise activities will be conducted two days per week [19] concurrently with the nutrition session, which is only once per week. The multicomponent exercise program is an adaptation with a slight modification of the frailty intervention study conducted by Tarazona-Santabalbina, which involves a progressive combination of activities including proprioception and balance exercises, aerobic training, resistance training, and stretching [29]. All exercise sessions will be delivered in group as it is effective in reducing or postponing frailty [46] and will be supervised by qualified physiotherapists [29]. For nutrition education intervention, five sessions of group discussion are conducted among participants; the main idea being to educate the elderly to improve their intake of calories, protein [47,48], calcium, and vitamin D [49]. Thus, every week they will receive an explanation about the importance of each nutrient, its sources, and the recommendation for daily intake. The session will also be interspersed with a few quizzes and hands-on activities, such as cooking practice [18] and games [50], to reinforce the learning objectives. In the following 12 weeks after the intervention ends, one session is added with the purpose of doing revision regarding the previous education session and to supervise the exercise routine practice by the elderly. The session will be conducted at week 6 after the intervention ends and will be delivered by a dietitian and a physiotherapist from the research team, lasting for one hour. The respondents in the control group will not receive any intervention program. A summary of the FINE program is shown in Table 2.

### 2.4. Outcome Measures

In this research study, the primary and secondary outcomes will focus on three key domains, which are physical changes, cognitive performance, and functional improvement. This includes the changes in frailty status and score, knowledge, attitude, and practice (KAP) towards frailty, nutrition and exercise, dietary intake, anthropometric measurements, cognitive status, functional ability status, mobility status, and risk of falls. Apart from that, demographic data (e.g., gender, race, age, monthly income, education level, and marital status) will be assessed as well.

#### 2.4.1. Primary Outcome

##### Frailty Score/Status

The frailty status will be assessed using the well-established Malay language standardized phenotype of the frailty questionnaire; as proposed by Fried et al. [5]. The questionnaire evaluates five components of the frailty syndrome (weight loss, exhaustion, weakness, slowness, and low activity) and allocates one point for each criterion met; respondents meeting zero criteria are defined as non-frail, whereas those meeting one or two criteria are defined as pre-frail, and those meeting three, four, or five criteria are defined as frail.

#### 2.4.2. Secondary Outcomes

##### Knowledge, Attitude, and Practice (KAP) towards Frailty, Nutrition, and Exercise

The KAP questions will be developed prior to data collection to measure the construct of knowledge, attitude, and practice in relation to frailty, nutrition, and exercise among the elderly. The questions will consist of items based on the content of the developed educational materials. A validation and reliability study will be conducted after it is developed.

##### Dietary Intake

The amount of food items consumed by the respondents in the past week will be recorded by the validated Malay language of Diet History Questionnaire (DHQ) via the interview method. The data will be analyzed using the Nutritionist Pro™ Diet Analysis Software and then compared to the Malaysian Recommended Nutrient intake [51].

##### Anthropometric Measurements

Body weight will be measured using an electronic flat scale (SECA 803, Hamburg, Germany).Height of the participant will be measured using stadiometer (SECA 213, Hamburg, Germany).When no valid measurement can be obtained due to particularities (e.g., kyphosis, scoliosis), an alternative method will be used to measure the height using a demi-span measurement. The height will be measured using equations developed for the Malaysian elderly [52].Body Mass Index (BMI) will be derived using the calculation: weight in kilograms divided by the height in meters squared [weight (kg)/height^2^ (m^2^)] [53]Body part circumference including mid-upper arm circumference (MUAC) and calf circumference (CC) will be measured using a flexible non-stretchable measuring tape (SECA 201, Hamburg, Germany) [52].Body composition including body fat percentage and muscle percentage will be assessed using a portable body composition analyzer (OMRON HBF-375, Japan).

##### Cognitive Status

This study will use the validated Malay language version of the Mini-Mental State Examination (M-MMSE) questionnaire to assess the cognitive functioning of the respondents [54].

##### Functional Ability

The well-established Malay language version of the Lawton Instrumental Activities of Daily Living (IADL) questionnaire will be used to assess the respondents’ ability to perform eight daily activities (i.e., ability to use telephone, shopping, preparing meals, housekeeping, doing laundry, using public transport, taking medications, and handling finances) [1,15].

##### Mobility Status

A well-established Malay language version of the Short Physical Performance Battery (SPPB) will be used for this study to test the respondents’ mobility status [55] to evaluate lower limb extremity functioning in three components (balance test, gait speed test, and repeated chair stand test).

##### Risk of Falls

The Malay language version of the Berg Balance Scale (BBS) will be used to assess the risk of falls among the respondents [56]. A process of back translation and monolingual testing will be conducted for the questionnaire prior to the evaluation process [57].

### 2.5. Statistical Analysis

The description of the respondents’ characteristics will be reported by group (control and intervention). Depending on the distribution of the variable of interest, the descriptive statistics of continuous data will be presented by using the mean and standard deviation and the median and the inter-quartile range. Categorical data will be presented as frequencies and percentages. For primary analysis, since there are small number of cluster, as recommended by Hayes and Moulton (2017), analysis will be conducted at cluster level instead of individual level [58]. Thus, the primary and secondary outcomes will be compared within and between groups using 2-way repeated ANOVA by general linear model. Any covariate data at baseline value will be adjusted and further test will be conducted using a repeated measure analysis of covariance (ANCOVA). If the data is not distributed normally, Kruskall–Wallis test will be used to analyze the data. Statistical significance is set at *p* < 0.05. Data will be entered and analyzed using SPSS 25.0 software.

Intention-to-treat (ITT) will be applied in the data management and analysis. All data will be processed according to their arm even if the participant did not manage to come to all the intervention sessions. It is assumed that majority of the participant in the intervention arm will receive adequate number of the intervention sessions. Action will be taken in order to prevent lower attendance from the participants such as conducting extra exercise classes for participants who could not come to a certain session. For every nutrition session, class will start with the revision on the topic that had been discussed in previous session. This can help participants who could not come to the previous nutrition session to learn the topic they missed.

## 3. Discussion

The FINE intervention program presents a unique approach, delivering a multi-domain intervention for pre-frail elderly men and women who live in the urban community. Often, research had been made among poor rural elderly, showing their high risk of getting malnutrition and becoming frail, but less is known among poor elderly population that live in urban areas such as the elderly community in PPR, Kuala Lumpur. Considering the high prevalence of the pre-frail elderly in urban areas [15], this target group should not be neglected. Aligned with the main objectives to improve the frailty status of the elderly, the program will have a highly positive impact in many ways, i.e., physically, mentally, and in social life.

The major strength of the proposed study design is the implementation of a multi-domain intervention that was considered to be more effective that a single intervention, especially through nutrition and exercise [33,34]. In addition, the designated nutritional education intervention was made based on the study that had demonstrated positive outcomes among frail elderly [18,26,59]. A few elderly-related local nutrition-education programs were also reviewed during the design process [1,60], considering the cultural values of local elderly population.

The strength of the nutrition education in this study mainly highlights the strategy to tackle the unintentional weight loss problem, which is also one of the main problems among frail elderly people [61]. Targeting one of the risk factors of frailty is also one of the key strategies for frailty prevention, as highlighted in the report of Asia Pacific CPGs for the management of frailty. The multicomponent exercises were applied in this study because of the strong evidence that performing different types of exercise assists in frailty prevention and improves the outcomes [35,36], and it is strongly recommended by Asia Pacific CPGs. The multi-component exercises were adapted from the exercise programs that were feasibly delivered to frail older people [29,32] with slight modifications to ensure better adherence to the exercise program by the local elderly.

An additional advantage of this community-based intervention is that it will be conducted in a natural setting, allowing respondents to integrate changes into their daily lives in a real-life context. This implies that the intervention delivered can be potentially transferable to other community settings. Moreover, it is also readily transferable to routine clinical practice in a health service setting, and the interdisciplinary approach is relevant to several professional groups in health care such as nutritionists, dietitians, and physiotherapists.

## 4. Conclusions

In overall, this FINE intervention program has been designed to be interactive, enjoyable, and manageable to encourage adherence among the respondents. Based on the evidence, we are convinced that this intervention program will be able to prevent frailty and the adverse effects associated with pre-frail elderly people. If the intervention produces significant positive effects, the findings will potentially provide valuable evidence and serve as a model of locally acceptable strategies to prevent frailty and reduce adverse health outcomes among older Malaysians.

## Figures and Tables

**Figure 1 nutrients-12-02758-f001:**
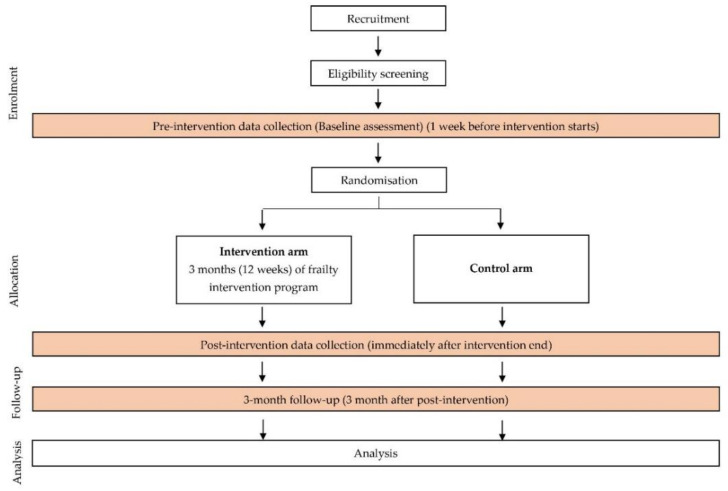
The overview of the frailty intervention study flow. Adapted from the CONSORT 2010 flow diagram (extension to cluster randomized trails) [41].

**Table 1 nutrients-12-02758-t001:** The summary of the inclusion and exclusion criteria for the Frailty Intervention through Nutrition Education and Exercise (FINE) program.

Inclusion Criteria	Exclusion Criteria
Men or women aged 60 years and above	Self-reported chronic diseases and mental illnesses (heart-related disease, chronic obstructive pulmonary disease (COPD), stroke, cancer, asthma, renal dysfunction, terminally ill, major depression, bipolar, disorder, obsessive compulsive disorder and post-traumatic disorder)
Meet one or two frailty phenotype score (pre-frail)	Physical Activity Readiness-Questionnaire (PAR-Q & YOU)(Yes ≥ 1) [43]
Able to ambulate without personal assistance	Bedridden
Residing in the PPR flats in Kuala Lumpur	Cognitive impairment (Elderly Cognitive Assessment Questionnaire, ECAQ < 6) [44]
Willing to participate in the intervention program with informed consent	Sensory impairment (visual & hearing) that will interfere with communication
	Unable to read and write
	Already involved or still participating in any health interventional study
	Any sustained fracture (hip, vertebrae) in past six months
	Any surgery (hip, abdominal area) in past six months

**Table 2 nutrients-12-02758-t002:** Summary of the frailty FINE program.

Intervention Group					Control Group
12 weeks intervention program	**Week**	**Day 1**	**Day 2**	**Day 3**	* No intervention
	1	Talk on frailty	Talk on exercise	Talk on healthy eating	
	2	Multicomponent exercise *			
	3	Multicomponent exercise *		Nutrition education class	
	4	Multicomponent exercise *			
	5	Multicomponent exercise *		Nutrition education class	
	6	Multicomponent exercise *			
	7	Multicomponent exercise *		Nutrition education class	
	8	Multicomponent exercise *			
	9	Multicomponent exercise *		Nutrition education class	
	10	Multicomponent exercise *			
	11	Multicomponent exercise *		Nutrition education class	
	12	INTERVENTION END			

* exercise intervention.

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
