# Peer review of "Frailty Intervention through Nutrition Education and Exercise (FINE). A Health Promotion Intervention to Prevent Frailty and Improve Frailty Status among Pre-Frail Elderly—A Study Protocol of a Cluster Randomized Controlled Trial"

_nutrients, 2020, doi:10.3390/nu12092758_

Round 1

Reviewer 1 Report

This is a well designed study, which aims to address frailty in older people living in an urban area of Malaysia. Some minor changes in syntax and the style of writing are required throughout. For example, in line 18 of the abstract you need to complete the sentence by stating what the risk is (i.e. frailty. poor clinical outcomes etc.). 

Author Response

Point 1:   Some minor changes in syntax and the style of writing are required throughout. For example, in line 18 of the abstract you need to complete the sentence by stating what the risk is (i.e. frailty. poor clinical outcomes etc.). 

Response 1: Line 18, the sentences, “elderlies are at high risk” has been removed since the sentences seems incomplete. Instead of list out the risks as suggested by the reviewer, we had changed the sentences for a better introduction to the intervention program. We didn’t make much changes of the style of writing. The paper has undergone a proofreading process for English language editing before. Thus, thorough changes of style of writing may need the service from expert again.

Point 2: Research design and method description can be improved.

Response 2: Both research design and method has been described in details. A clear explanation of the screening process also has been added (line 128-145).

Reviewer 2 Report

The topic discussed by the authors is a very important element of caring for the health of the elderly. It is a project definitely worth further development.

Line 126-131. From how many flats will the sample of 31 flats be selected? Where does the number 31 come from? Do the authors mean apartments in blocks of flats? The description of the flats is not entirely clear.

Table 1. Do the authors take into account mental illness and dementia?

Author Response

Point 1:Is the research design appropriate? – can be improved. Line 126-131. From how many flats will the sample 31 flats be selected? Where does the number 31 come from? Do the authors mean apartments in blocks of flats? The description of the flats is not entirely clear. 

Response 1: An explanation was added to explain where the number 31 come from. In total there are 32 PPR flats listed under Kuala Lumpur area (DBKL, 2019). 1 PPR flats is excluded for the purpose of acceptability study as part of the development process of the education materials. The remaining 31 flats will be included for further study (screening and intervention). Meanwhile, the description of the flats has been described in brief in the introduction but not include the details e.g. list number of block or number of units for each flat because each flat has different number of block and unit. Plus, the list mentioned above only been used for the process of proportionate sampling method for screening study. Since, the focus of the paper is about the intervention program, thus, we decided to not include the list mentioned above.

However, since the screening study also important as prerequisite of the intervention program, I also added the summary on the explanation about the screening study (e.g. type of study design and sampling procedure, number of sample and measurements) (line 129-146)

Point 2: Are the results clearly presented? Are the conclusions supported by the results? - can be improved.

Response 2:In this protocol paper, there is no any result that are presented since the actual data collection is not started yet. Since, there is no data, the conclusion also was made only based on the discussion.

Point 3: Table 1. Do the authors take into account mental illness and dementia?

Response 3: Mental illnesses can disrupt a person’s thinking, feeling, mood, ability to socialize with others and daily functioning. It is often results in reduced the ability to cope with the routine daily activities (Centers for Disease Control and Prevention, 2012). Since this might affect the result of the intervention program, participant who has self-reported of mental illness such as major depression, bipolar disorder, obsessive compulsive disorder and post-traumatic stress disorder also will be excluded from this study. This statement has been added in the table 1 under the exclusion criteria. 

Meanwhile, for dementia, it is being concerned through the Elderly Cognitive Assessment Questionnaire (ECAQ) score where score of 5 and less indicates cognitive impairment with “probable dementia” (Sherina et al., 2004) and will be excluded from this study

Round 2

Reviewer 1 Report

Thanks for the additional clarification. Wishing you all the best with the project.